**Data Availability Statement:** The dataset used in this study is held by the Taiwan Ministry of Health and Welfare (MOHW). The Ministry of Health and

# Drug-eluting versus bare-metal stents for first myocardial infarction in patients with atrial fibrillation: A nationwide population-based cohort study

Nen-Chung Chang[1], Patrick Hu[2,3], Tien-Hsing Chen[4], Chun-Tai Mao[4], Ming-Jui Hung[4], Chi-Tai Yeh[5,6], Ming-Yow Hung[7,8]*

1 Division of Cardiology, Department of Internal Medicine, Taipei Medical University Hospital, Taipei, Taiwan, 2 University of California, Riverside, Riverside, California, United States of America, 3 Department of Cardiology, Riverside Medical Clinic, Riverside, California, United States of America, 4 Division of Cardiology, Department of Medicine and Community Medicine Research Center, Chang Gung Memorial Hospital, Keelung, Chang Gung University College of Medicine, Keelung City, Taiwan, 5 Department of Medical Research and Education, Shuang Ho Hospital, Taipei Medical University, New Taipei City, Taiwan, 6 Department of Medical Laboratory Science and Biotechnology, Yuanpei University of Medical Technology, Hsinchu City, Taiwan, 7 Division of Cardiology, Department of Internal Medicine, Shuang Ho Hospital, Taipei Medical University, New Taipei City, Taiwan, 8 Department of Internal Medicine, School of Medicine, College of Medicine, Taipei Medical University, Taipei City, Taiwan

* myhung6@ms77.hinet.net

## Abstract

### Background

Acute myocardial infarction (AMI) complicates the clinical management of atrial fibrillation (AF) because coronary stenting may influence subsequent antithrombotic therapy. We investigated the use of a bare-metal stent (BMS) or a drug-eluting stent (DES) and associated outcomes in patients with pre-existing AF and first AMI undergoing percutaneous coronary intervention.

### Methods and results

Patient records in this population-based study were retrospectively collected from the Taiwan National Health Insurance Research Database. Using propensity score matching (PSM), we used 1:2 ratio stratification into a DES group of 436 and a BMS group of 785 patients from 2007 to 2011. The mean follow-up of matched cohorts was 1.7 years. After PSM, DESs were associated with lower rates of cardiovascular death (7.8%, hazard ratio [HR] 0.58, 95% confidence interval [CI] 0.39–0.86 and 10.1%, HR 0.64, 95% CI 0.45–0.90) and primary composite outcome (35.1%, HR 0.76, 95% CI 0.63–0.92 and 48.2%, HR 0.81, 95% CI 0.69–0.96) than BMSs within the first year and at the end of follow-up. Although the greatest benefit from DESs, irrespective of the first- and second- generation DESs, implantation was observed within the first year only, this benefit was not observed in patients with diabetes, chronic kidney disease, or dialysis.

Welfare must approve an application to access this data. Any researcher interested in accessing this dataset can submit an application form to the Ministry of Health and Welfare requesting access. Please contact the staff of MOHW (Email: stcarolwu@mohw.gov.tw) for further assistance. Taiwan Ministry of Health and Welfare Address: No.488, Sec. 6, Zhongxiao E. Rd., Nangang Dist., Taipei City 115, Taiwan (R.O.C.). Phone: +886-2-8590-6848.

**Funding:** This work was supported by National Science Council of Taiwan grant to Ming-Yow Hung (MOST-108-2314-B-038-119-MY3); and Taipei Medical University–Shuang Ho Hospital (106TMU-SHH-06) to Ming-Yow Hung. The funders had no role in study design, data collection and analysis, decision to publish, or preparation of the manuscript.

**Competing interests:** The authors have declared that no competing interests exist.

## Conclusions

Use of DESs in AMI patients with pre-existing AF is associated with significantly lower rates of cardiovascular death and primary composite outcome within the first year follow-up. However, the effect is not apparent in patients with diabetes, chronic kidney disease or dialysis.

## Introduction

Atrial fibrillation (AF), the most common sustained arrhythmia, is an independent predictor of acute myocardial infarction (AMI) [1]. While 9.3% of acute coronary syndrome patients have a history of AF [2], the annual rate of AMI in patients with pre-existing AF (AF patients) ranges from 0.4% to 2.5% [3]. Furthermore, a lower annual AMI rate has been described in Asian people (0.2–0.3%) [4], suggesting racial heterogeneity in the development of AF-related AMI and underscoring the importance of evaluating the racial differences in associated outcomes. In patients with coronary artery disease undergoing percutaneous coronary intervention (PCI) and stent implantation, the intracoronary and systemic prothrombotic environment accompanying an acute coronary syndrome rather than stable angina has led to concerns regarding a possible higher risk of stent thrombosis [5].

Consequently, the controversial role of drug-eluting stent (DES) during PCI for MI [6] and the lower use of DES than bare-metal stent (BMS) in AMI patients reflects contemporary clinical practice. On the other hand, in elderly patients with AF undergoing PCI, the use of triple therapy, including dual-antiplatelet therapy (DAPT) and an oral anticoagulant, compared to DAPT alone was associated with reduced thromboembolism and mortality rates, although a higher rate of major bleeding [7]. While the bleeding risk of AF patients is increased due to triple therapy, the most appropriate strategy to balance thrombotic complications after PCI for AF patients requiring coronary stent implantation, are unclear. In the United States, authors of 2006 AF guidelines suggest that the most important agent for the maintenance of stent patency is the thienopyridine derivative clopidogrel and that the addition of aspirin to the chronic anticoagulant regimen contributes more risk than benefit [8]. The use of clopidogrel without aspirin is associated with a reduction in bleeding and no increase in the rate of stent thrombotic events [9]. Moreover, the omission of aspirin while maintaining clopidogrel and oral anticoagulant has been evaluated in the WOEST trial, in which 573 anticoagulated patients undergoing PCI (70% with AF) were randomized to either dual therapy with oral anticoagulant and clopidogrel (75 mg once daily) or to triple therapy with oral anticoagulant, clopidogrel, and aspirin [10, 11]. Bleeding was lower in the dual vs. triple therapy arm, driven by fewer minor bleeding events. The rates of myocardial infarction, stroke, target vessel revascularization, and stent thrombosis did not differ (albeit with low event numbers), but all-cause mortality was lower in the dual therapy group at 1 year (2.5% vs. triple therapy 6.4%). Although the trial was too small to assess ischemic outcomes, dual therapy with oral anticoagulant and clopidogrel but without aspirin may emerge in the future as an alternative to triple therapy in patients with AF and ACS and/or PCI [10, 12, 13]. Similar results of 2 randomized clinical trials, PIONEER AF-PCI and RE-DUAL PCI, support the concept that an oral anticoagulant in combination with single antiplatelet therapy without aspirin, a strategy known as double antithrombotic therapy, is superior to a strategy of triple therapy consisting of the combination of an oral anticoagulant and DAPT in reducing bleeding complications [14–16]. On the other hand, the latest 2016 European AF guideline provides recommendations only on

antithrombotic therapy strategy, not stent selection [10], which leaves a critical clinical dilemma in the selection of stents when a period of DAPT is also required to minimize the risk of stent thrombosis.

There is significant heterogeneity in the professional guidance regarding the best type of stent to implant in patients with pre-existing AF. Before 2009, the use of DES in AF patients have not been tested in clinical trials to study the efficacy and safety of these stents, with AF being a common exclusion criterion in clinical trials of DES. In the United States, a class 3 recommendation (harm) appeared in 2013 ST-segment elevation MI guidelines for the use of DESs in patients unable to comply with long-term DAPT [17], which may be a consideration in patients requiring long-term oral anticoagulants. While the first 2011 North American consensus document specifically recommends avoiding DES use in patients with AF and high bleeding risk [18], the use of BMS is considered to minimize the duration of DAPT (class 2b, level of evidence C) in AF guidelines [9, 19]. However, in parallel with the advances in antiplatelet therapy, the second 2016 North American consensus document suggests that stents have become safer, with new-generation DES having a lower rate of stent thrombosis than the first-generation DES, and even potentially lower rates than with BMS across manifestations of coronary artery disease, including those with acute ST-segment-elevation MI [20]. In Europe, guidance that is more recent suggests that current-generation DESs may be preferred [21]. However, scarce data are available regarding stent selection and associated outcomes in AF patients presenting with first AMI undergoing PCI. Among AF patients, women are at higher risk of MI than men are, as shown in the REGARDS study [1]. On the other hand, women have smaller coronary arteries and are treated less often with DES than men [22], leading to inferior results following PCI [23]. We, therefore, examined the characteristics of AF patients with a first AMI who received a BMS or a DES, and the effect of BMS versus DES, on interventional outcomes in this nationwide population-based cohort.

## Materials and methods

### Data source

We designed this observational prospective cohort study using the retrospective collected claims data of AF patients presenting with a first episode of AMI from the Taiwan National Health Insurance Research Database (NHIRD) between 2007 and 2011. Disease was detected using the International Classification of Diseases, Ninth Revision, Clinical Modification (ICD-9-CM) codes in the NHIRD. The NHIRD is generated from the National Health Insurance (NHI) system, a government-operated compulsory health insurance system providing medical care to higher than 99.8% of the 23 million Taiwanese people, and reimburses all the medical expenditures with very few exceptions. The accuracy of all insurance claims was audited under a peer review system conducted by several government-appointed medical specialists. Further information regarding NHI and NHIRD have been described in previous publications [24–26].

### Ethics statement

The encryption system used in the NHIRD makes identifying individuals impossible. Confidentiality is assured by abiding by the Bureau of NHI regulations for data retrieval and use. Therefore, Chang Gung Memorial Hospital Institutional Review Board approval was waived due to the secondary nature of the de-identified data in the retrospective study design. The Chang Gung Memorial Hospital Institutional Review Board specifically waived the consent requirement to access the NHIRD.

## Study population

This study identified adult patients ($\geq$20 years) admitted with a first episode of AMI (ICD-9-CM code: 410.xx) from January 1, 2007, to December 31, 2011, from the entire Taiwan population. We chose the first AMI admission as the index hospitalization during the study period when one patient had $\geq$2 AMI admissions. The positive predictive value of diagnosis codes of AMI was 93% in a previous NHIRD study [27]. We assigned the discharge dates for the first AMI as the index dates. The exclusion criteria were as the follows: prior coronary stent implantation, PCI without stenting during the index hospitalization, or PCI with both BMS and DES during the index hospitalization. The patients were followed from the index date until the date of event occurrence, the date of death or December 31, 2011, whichever came earlier [6, 28].

## Exposure

The exposure of primary interest in this study was the type of stent, namely DES versus BMS, which was used during the index PCI procedure. The type of stent was extracted using Taiwan NHI reimbursement codes in the inpatient claims data. DESs included the first-generation (sirolimus-eluting, paclitaxel-eluting sternts), and the second-generation DESs (everolimus-eluting, zotarolimus-eluting, biolimus-eluting and tacrolimus-eluting stents).

## Covariate

Covariates were age, sex, monthly income, urbanization level, hospital level of the index AMI hospitalization, 14 comorbidities, 2 previous cardiac interventions (PCI or coronary artery bypass grafting), $CHA_2DS_2$-VASc score, number of intervened diseased vessels, number of stents implanted per patient, in-hospital complications and procedures, 15 kinds of medication at discharge and admission duration. Previous cardiac interventions and in-hospital procedures were retrieved using Taiwan NHI reimbursement codes. The comorbidities were detected using an inpatient diagnosis before the index date, which could be tracked up to year 1997. The ICD-9-CM diagnostic codes of the selected comorbidities were provided in the supplement (S1 Table) [28]. The $CHA_2DS_2$-VASc score at the index hospitalization was calculated. We also extracted the prescription records of relevant medications at discharge from the inpatient claims data.

## Outcome

The primary composite outcome was anyone of ischemic stroke, AMI, revascularization and all-cause mortality. The accuracy of the diagnosis codes for ischemic stroke have been verified in previous NHIRD studies with positive predictive values $\geq$95% [27, 29]. Revascularization by PCI or coronary artery bypass grafting was extracted using the Taiwan NHI reimbursement codes [6]. Due to the obligational and mandatory nature of Taiwan NHI system, a withdrawal from the insurance system was considered a death [30]. We defined cardiovascular death according to the criteria of the Standardized Definitions for Cardiovascular and Stroke End-point Events in Clinical Trials by the US Food and Drug Administration [31]. Secondary cardiac and safety outcomes were defined as being admitted with a principal diagnosis of any stroke, hemorrhagic stroke, unspecified stroke, gastrointestinal bleeding, major bleeding and heart failure hospitalization [6, 28, 32].

## Statistical analysis

We used propensity score matching to reduce potential confounding effects on the observed variables and selection bias before comparing clinical outcomes between groups (DES versus

BMS). The propensity score was the predicted probability to be in the DES group given values of covariates as determined using logistic regression. Selected covariates included 5 demographics (sex, age, monthly income, urbanization level and hospital level), 14 comorbidities, CHA2DS2-VASc score, prior coronary treatment (PCI or coronary artery bypass grafting), angiographic and procedural characteristics, medications administered at discharge, intensive care unit duration, admission durations and the admission date. The matching was processed using a greedy nearest neighbor algorithm with a caliper of 0.2 times of the standard deviation of the logit of the propensity score, with random matching order and without replacement [28]. We matched each patient in the DES group with 2 patients (if possible) in the BMS group.

Data of categorical and continuous variables are presented as number and percentages and mean ± standard deviation, respectively. We compared the baseline characteristics of patients between groups by the chi-square test for categorical variable or by 2-sample *t* test for continuous variable. The risks of time-to-event outcomes between the groups were compared by the Cox proportional hazard model. The study group (DES versus BMS) was the only explanatory variable in the Cox regression. The matched pairs were stratified in the Cox model to account for the outcome dependency within the same matching pair due to matching. We performed a pre-specified subgroup analysis of the primary composite outcome to investigate whether the observed treatment effect on primary composite outcome was consistent across different levels of subgroups. The selected subgroup variables were gender, age, hypertension, diabetes, chronic kidney disease, dialysis, old MI, old stroke, different CHA2DS2-VASc score, shock during admission, acute kidney injury during admission, warfarin, amiodarone, ACEI/ARB, statin and PPIs. Finally, we compared the risks of revascularization, cardiovascular death and primary composite outcome among different stent types (BMS, first- and second- generation DESs) by pairwise log-rank test with Bonferroni adjustment.

A 2-sided P value <0.05 was considered statistically significant. We made no multiple testing (multiplicity) adjustments in this study. The P value for interaction represents the likelihood of interaction between the variable and the treatment effect (DES versus BMS). We performed all statistical analyses using a commercial software (SAS 9.4, SAS Institute, Cary, NC), including the procedures of 'psmatch' for propensity score matching and 'phreg' for survival analysis.

## Results

### Patient characteristics

A total of 88,404 patients firstly admitted with a principal diagnosis of AMI between January 2007 and December 2011 were identified. Among these patients, 8,597 (9.7%) had a history of AF. We further identified 1,971 AF patients who were admitted for first AMI and who subsequently received coronary stenting (Fig 1). Of those, 1,528 patients (77.5%) underwent BMS implantation and 443 (22.5%) underwent DES implantation. Upon propensity score matching, 349 and 87 DES-treated patients had 2 and 1 counterparts, respectively, resulting in 436 patients in the DES group and 785 patients in the BMS group. The mean follow-up of the matched cohort was 1.7 years (standard deviation = 1.4 years).

The mean age of patients was 73.2 ± 11.5 years and nearly 70% were men. Before propensity score matching was done, DES-treated patients were more likely to live in urbanized area, to receive PCI in a community hospital (not a major medical center), had a lower prevalence of heart failure, chronic obstructive pulmonary disease, stroke, and major bleeding. The DES patients were also more likely to have undergone a prior PCI, had lower CHA2DS2-VASc scores, a higher number of treated vessels, and were less likely to have undergone intra-aortic

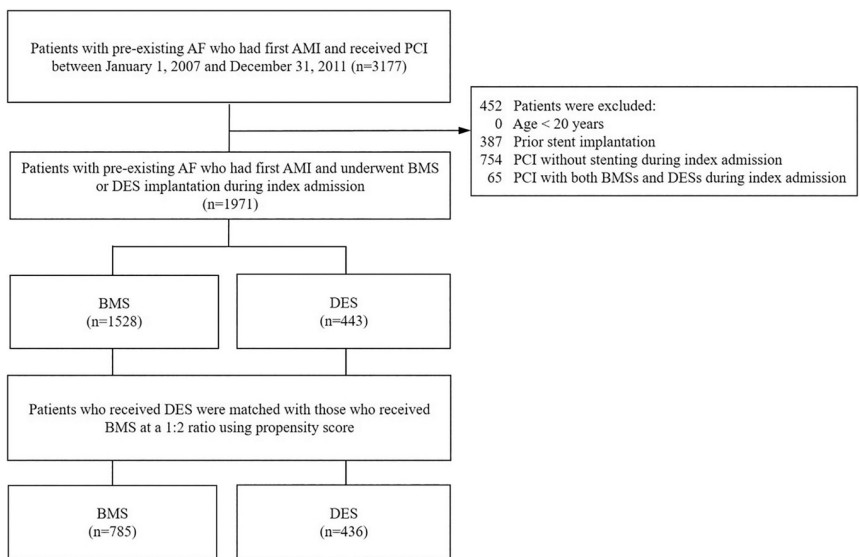

**Fig 1. Flow chart for study patient inclusion.** AF, atrial fibrillation; AMI, acute myocardial infarction; BMS, bare-metal stent; DES, drug-eluting stent; PCI, percutaneous coronary intervention.

balloon pump insertion and intubation. They were also less likely to develop cardiogenic shock and receive digoxin and proton-pump inhibitors and were more likely to receive oral hypoglycemic agents, beta-blockers, angiotensin converting enzyme inhibitor/angiotensin II receptor blockers, dihydropyridine calcium channel blockers and statins (P<0.05). After propensity score matching, the baseline characteristics of the 2 groups were well-balanced with insignificant differences in terms of all variables (Table 1).

### Clinical outcomes

No significant differences in the risks of ischemic stroke, MI and revascularization during the 1-year follow-up or the end of follow-up were observed. However, the event rate of cardiovascular death during the 1-year follow-up was 7.8% (34/436) and 13.0% (102/785) in the DES and BMS groups, respectively (Table 2). The DES patients had significantly lower risks of cardiovascular death (hazard ratio [HR] 0.58, 95% confidence interval [CI] 0.39–0.86) and primary composite outcome (HR 0.76, 95% CI 0.63–0.92). Moreover, the event rates of the primary composite outcome during the 1-year follow-up were 35.1% (153/436) and 43.3% (340/785) in the DES and BMS groups, respectively.

At the end of follow-up, the event rate of cardiovascular deaths was 10.1% (44/436) and 15.0% (118/785) in the DES and BMS groups, respectively. The DES patients had significantly lower risks of cardiovascular death (HR 0.64, 95% CI 0.45–0.90; Fig 2A) and primary composite outcome (HR 0.81, 95% CI 0.69–0.96; Fig 2C). The event rates of primary composite outcome were 48.2% (210/436) and 54.4% (427/785) in the DES and BMS groups, respectively (Table 2). Although the benefit from DES implantation was the greatest only during the first year after treatment for first AMI, DES or BMS selection demonstrated insignificant differences of cardiovascular death (Fig 2B) and primary composite outcome (Fig 2D) beyond the first year follow-up.

Regarding secondary and safety outcomes, we found no significant differences in the risks of stroke (HR 0.93, 95% CI 0.62–1.38), heart failure admission (HR 0.72, 95% CI 0.50–1.02), or major bleeding (HR 0.85, 95% CI 0.51–1.41) between groups (Table 3).

**Table 1. Baseline characteristics of patients before and after propensity score matching.**

| Characteristics | Before matching | | | After matching | | |
|---|---|---|---|---|---|---|
| | BMS | DES | P | BMS | DES | P |
| Patient number | 1,528 | 443 | — | 785 | 436 | — |
| Age (year) | 73.5 ± 11.6 | 72.3 ± 11.0 | 0.061 | 72.6 ± 11.6 | 72.3 ± 11.0 | 0.696 |
| Age ≥65 years | 1193 (78.1) | 342 (77.2) | 0.696 | 592 (75.4) | 336 (77.1) | 0.518 |
| Male | 1,027 (67.2) | 308 (69.5) | 0.359 | 545 (69.4) | 303 (69.5) | 0.980 |
| Monthly income–NTD$ | | | 0.050 | | | 0.641 |
| 0 | 500 (32.7) | 147 (33.2) | | 264 (33.6) | 145 (33.3) | |
| 1–20000 | 432 (28.3) | 101 (22.8) | | 198 (25.2) | 101 (23.2) | |
| > 20000 | 596 (39.0) | 195 (44.0) | | 323 (41.1) | 190 (43.6) | |
| Urbanization level | | | <0.001 | | | 0.675 |
| 1–most urbanized | 353 (23.1) | 125 (28.2) | | 222 (28.3) | 123 (28.2) | |
| 2 | 444 (29.1) | 159 (35.9) | | 258 (32.9) | 155 (35.6) | |
| 3 | 511 (33.4) | 116 (26.2) | | 213 (27.1) | 115 (26.4) | |
| 4–least urbanized | 220 (14.4) | 43 (9.7) | | 92 (11.7) | 43 (9.9) | |
| Hospital level | | | <0.001 | | | 0.372 |
| Medical center (teaching hospital) | 821 (53.7) | 179 (40.4) | | 343 (43.7) | 179 (41.1) | |
| Regional / district hospital | 707 (46.3) | 264 (59.6) | | 442 (56.3) | 257 (58.9) | |
| Comorbidities | | | | | | |
| Hypertension | 1,043 (68.3) | 303 (68.4) | 0.956 | 546 (69.6) | 298 (68.3) | 0.662 |
| Diabetes mellitus | 587 (38.4) | 185 (41.8) | 0.204 | 318 (40.5) | 181 (41.5) | 0.732 |
| Dyslipidemia | 395 (25.9) | 127 (28.7) | 0.237 | 232 (29.6) | 125 (28.7) | 0.745 |
| Heart failure | 403 (26.4) | 84 (19.0) | 0.001 | 168 (21.4) | 84 (19.3) | 0.377 |
| Chronic kidney disease | 121 (7.9) | 25 (5.6) | 0.107 | 52 (6.6) | 25 (5.7) | 0.540 |
| Dialysis | 85 (5.6) | 20 (4.5) | 0.387 | 45 (5.7) | 20 (4.6) | 0.393 |
| Gout | 193 (12.6) | 43 (9.7) | 0.095 | 75 (9.6) | 41 (9.4) | 0.932 |
| Chronic obstructive pulmonary disease | 352 (23.0) | 68 (15.3) | 0.001 | 133 (16.9) | 67 (15.4) | 0.476 |
| Peripheral arterial disease | 121 (7.9) | 31 (7.0) | 0.522 | 60 (7.6) | 30 (6.9) | 0.625 |
| Malignancy | 119 (7.8) | 29 (6.5) | 0.383 | 48 (6.1) | 28 (6.4) | 0.831 |
| Old myocardial infarction | 220 (14.4) | 76 (17.2) | 0.153 | 133 (16.9) | 74 (17.0) | 0.989 |
| Stroke | 389 (25.5) | 87 (19.6) | 0.012 | 169 (21.5) | 86 (19.7) | 0.457 |
| Gastrointestinal bleeding | 395 (25.9) | 95 (21.4) | 0.059 | 183 (23.3) | 92 (21.1) | 0.375 |
| Major bleeding | 160 (10.5) | 31 (7.0) | 0.030 | 57 (7.3) | 31 (7.1) | 0.922 |
| Previous treatment | | | | | | |
| Percutaneous coronary intervention | 122 (8.0) | 54 (12.2) | 0.006 | 87 (11.1) | 51 (11.7) | 0.745 |
| Coronary artery bypass grafting | 37 (2.4) | 12 (2.7) | 0.732 | 21 (2.7) | 12 (2.8) | 0.937 |
| $CHA_2DS_2$-VASc score | 4.0±2.2 | 3.8±2.1 | 0.050 | 3.9±2.1 | 3.8±2.2 | 0.608 |
| $CHA_2DS_2$-VASc score group | | | 0.037 | | | 0.987 |
| 1 | 221 (14.5) | 64 (14.4) | | 111 (14.1) | 64 (14.7) | |
| 2 | 193 (12.6) | 65 (14.7) | | 119 (15.2) | 64 (14.7) | |
| 3–5 | 688 (45.0) | 220 (49.7) | | 384 (48.9) | 215 (49.3) | |
| ≥ 6 | 426 (27.9) | 94 (21.2) | | 171 (21.8) | 93 (21.3) | |
| No. of intervened disease vessels | | | 0.003 | | | 0.487 |
| 1 | 1,203 (78.7) | 319 (72.0) | | 595 (75.8) | 317 (72.7) | |
| 2 | 290 (19.0) | 104 (23.5) | | 164 (20.9) | 102 (23.4) | |
| 3 | 35 (2.3) | 20 (4.5) | | 26 (3.3) | 17 (3.9) | |
| No. of stents implanted per patient | | | 0.558 | | | 0.870 |
| 1 | 1,076 (70.4) | 324 (73.1) | | 579 (73.8) | 317 (72.7) | |

(*Continued*)

**Table 1.** (Continued)

| Characteristics | Before matching | | | After matching | | |
|---|---|---|---|---|---|---|
| | BMS | DES | P | BMS | DES | P |
| 2 | 339 (22.2) | 89 (20.1) | | 155 (19.7) | 89 (20.4) | |
| 3 | 85 (5.6) | 25 (5.6) | | 39 (5.0) | 25 (5.7) | |
| 4 or more | 28 (1.8) | 5 (1.1) | | 12 (1.5) | 5 (1.1) | |
| Aspiration catheter used | 216 (14.1) | 52 (11.7) | 0.195 | 94 (12.0) | 52 (11.9) | 0.980 |
| Intra-aortic balloon pump | 222 (14.5) | 37 (8.4) | 0.001 | 73 (9.3) | 37 (8.5) | 0.634 |
| Intubation | 264 (17.3) | 46 (10.4) | <0.001 | 88 (11.2) | 45 (10.3) | 0.633 |
| Extracorporeal membrane oxygenation | 17 (1.1) | 6 (1.4) | 0.676 | 15 (1.9) | 6 (1.4) | 0.491 |
| Cardiogenic shock with MCS | 224 (14.7) | 37 (8.4) | 0.001 | 75 (9.6) | 37 (8.5) | 0.536 |
| Acute kidney injury | 74 (4.8) | 12 (2.7) | 0.053 | 30 (3.8) | 11 (2.5) | 0.227 |
| Stay of intensive care unit (days) | 5.5 ± 8.0 | 4.7 ± 8.4 | 0.059 | 4.7 ± 7.0 | 4.5 ± 7.4 | 0.635 |
| Medication at discharge | | | | | | |
| Digoxin | 304 (19.9) | 64 (14.4) | 0.010 | 138 (17.6) | 64 (14.7) | 0.191 |
| Warfarin | 119 (7.8) | 40 (9.0) | 0.398 | 64 (8.2) | 39 (8.9) | 0.633 |
| Amiodarone | 856 (56.0) | 229 (51.7) | 0.107 | 412 (52.5) | 226 (51.8) | 0.828 |
| Oral hypoglycemic agent | 378 (24.7) | 131 (29.6) | 0.041 | 226 (28.8) | 128 (29.4) | 0.834 |
| Insulin | 424 (27.7) | 110 (24.8) | 0.224 | 203 (25.9) | 109 (25.0) | 0.741 |
| Aspirin | 1,422 (93.1) | 422 (95.3) | 0.097 | 742 (94.5) | 415 (95.2) | 0.619 |
| Clopidogrel | 1,510 (98.8) | 439 (99.1) | 0.628 | 779 (99.2) | 432 (99.1) | 0.776 |
| Dual antiplatelet | 1,416 (92.7) | 420 (94.8) | 0.117 | 740 (94.3) | 413 (94.7) | 0.739 |
| Beta-blocker | 916 (59.9) | 310 (70.0) | <0.001 | 546 (69.6) | 304 (69.7) | 0.950 |
| ACEI / ARB | 1,108 (72.5) | 354 (79.9) | 0.002 | 621 (79.1) | 348 (79.8) | 0.770 |
| DCCB | 282 (18.5) | 110 (24.8) | 0.003 | 432 (55.0) | 257 (58.9) | 0.186 |
| Statin | 722 (47.3) | 260 (58.7) | <0.001 | 98 (12.5) | 50 (11.5) | 0.602 |
| Proton-pump inhibitor | 231 (15.1) | 50 (11.3) | 0.042 | 182 (23.2) | 104 (23.9) | 0.792 |
| NDCCB | 339 (22.2) | 86 (19.4) | 0.211 | 159 (20.3) | 85 (19.5) | 0.751 |
| Glycoprotein IIb/IIIa | 38 (2.5) | 9 (2.0) | 0.580 | 17 (2.2) | 9 (2.1) | 0.906 |
| Index admission duration (day) | 11.8 ± 15.9 | 10.3 ± 13.7 | 0.075 | 10.5 ± 12.8 | 10.1 ± 12.7 | 0.557 |
| Follow up years | 1.5 ± 1.4 | 1.8 ± 1.4 | 0.001 | 1.6 ± 1.4 | 1.8 ± 1.4 | 0.017 |

Values are means ± standard deviation, or numbers (percentages).

BMS, bare-metal stent; DES, drug-eluting stent; NTD, New Taiwan dollar; MCS, mechanical circulation support; ACEI, angiotensin converting enzyme inhibitor; ARB, angiotensin II receptor blocker; DCCB, dihydropyridine calcium channel blocker; NDCCB, non-dihydropyridine calcium channel blocker.

## Subgroup analyses

We further analyzed the primary composite outcome at the end of follow-up, stratified by patient's characteristics. The beneficial effect of DES was less apparent in patients with diabetes (P interaction = 0.0501), CKD (p interaction = 0.046), or dialysis (P interaction = 0.021) (Fig 3). Moreover, while no difference was found in the rates of revascularization associated with DES or BMS (Fig 4A), both the first- and second- generation DESs were associated with significantly lower rates of cardiovascular death (Fig 4B) and primary composite outcome (Fig 4C) than BMSs.

## Discussion

We found that among AF patients with a first AMI, the use of DESs, including both the first- and second- generation DESs, was associated with lower rates of cardiovascular death and

**Table 2. Cardiovascular outcomes.**

| Follow up / Outcome | Number of event (%) | | DES versus BMS HR (95% CI) | P |
|---|---|---|---|---|
| | BMS (n = 785) | DES (n = 436) | | |
| 1 year follow up | | | | |
| Ischemic stroke | 38 (4.8) | 25 (5.7) | 1.10 (0.66–1.82) | 0.716 |
| Myocardial infarction | 35 (4.5) | 19 (4.4) | 0.91 (0.52–1.59) | 0.745 |
| Revascularization | | | | |
| PCI | 143 (18.2) | 73 (16.7) | 0.84 (0.64–1.12) | 0.235 |
| CABG | 9 (1.1) | 2 (0.5) | 0.37 (0.08–1.70) | 0.201 |
| Death from any cause | 187 (23.8) | 67 (15.4) | 0.61 (0.46–0.81) | 0.001 |
| Cardiovascular death | 102 (13.0) | 34 (7.8) | 0.58 (0.39–0.86) | 0.006 |
| Primary composite outcome* | 340 (43.3) | 153 (35.1) | 0.76 (0.63–0.92) | 0.005 |
| At the end of follow-up | | | | |
| Ischemic stroke | 58 (7.4) | 37 (8.5) | 1.05 (0.69–1.58) | 0.835 |
| Myocardial infarction | 43 (5.5) | 32 (7.3) | 1.23 (0.78–1.94) | 0.385 |
| Revascularization | | | | |
| PCI | 178 (22.7) | 101 (23.2) | 0.92 (0.72–1.17) | 0.474 |
| CABG | 12 (1.5) | 4 (0.9) | 0.54 (0.17–1.66) | 0.279 |
| Death from any cause | 251 (32.0) | 107 (24.5) | 0.71 (0.57–0.89) | 0.003 |
| Cardiovascular death | 118 (15.0) | 44 (10.1) | 0.64 (0.45–0.90) | 0.011 |
| Primary composite outcome* | 427 (54.4) | 210 (48.2) | 0.81 (0.69–0.96) | 0.015 |

BMS, bare-metal stent; DES, drug-eluting stent; HR, hazard ratio; CI, confidence interval; PCI, percutaneous coronary intervention; CABG, coronary artery bypass graft; CV, cardiovascular.

*Any one of the following: ischemic stroke, myocardial infarction, revascularization, and death from any cause

primary composite outcome than BMSs within the first year and at the end of follow-up. While the greatest benefit from DES implantation was observed only within the first year of treatment, the outcomes were comparable upon prolonged follow-up. Regardless of the follow-up period, the cardiovascular benefits of DES were less apparent than BMS in patients with diabetes, CKD, or dialysis. Our study is the first to show an advantage in cardiovascular outcomes for DES compared to BMS in AF patients with first AMI requiring PCI.

In early trials of first-generation DES, the protocol-recommended DAPT duration was 3 to 6 months, but as a result of concerns about late thrombotic events, this was increased to 12 months in studies initiated after 2006 [33]. Coinciding with this shift, elderly patients or patients considered to be at high bleeding risk, high thrombotic risk, or low restenosis risk were largely excluded from the pivotal DES trials [33]. For the broad population, considering in-stent restenosis requiring revascularization, DESs are superior to BMSs within the first year of treatment [34]. Late in-stent restenosis occurs at approximately 1–2% per year with all types of stents and is similar for first- and second-generation DESs and BMSs [34]. On the other hand, although the overall risk of stent thrombosis at up to 1 year is low and is comparable for both DESs and BMSs as long as patients are continued on DAPT for the recommended duration [35], late and very late stent thrombosis after first-generation DES implantation needs long-term DAPT [36]. However, the long-term cumulative rate of stent thrombosis in acute coronary syndrome patients does not differ between first generation DESs and BMSs (5.8% versus 4.3%, respectively) [37]. Collectively, the rates of the 2 major causes of stent failure, restenosis and thrombosis, appear to be comparable between DESs and BMSs beyond 1 year after treatment. The interplay between the improved thrombogenicity of DESs and the ability

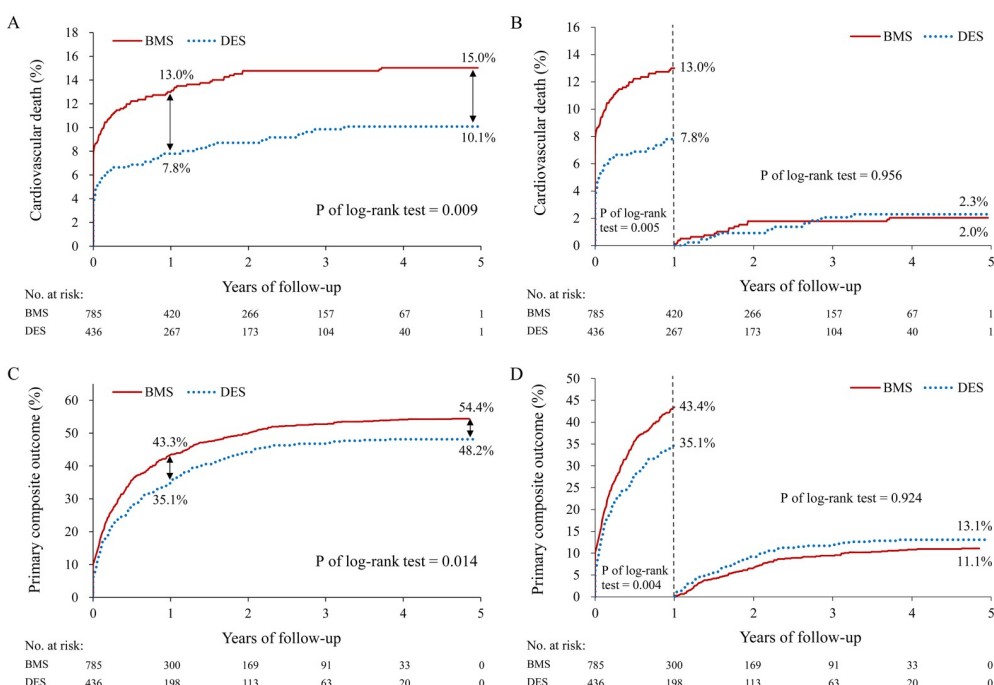

**Fig 2. Cumulative event rate of cardiovascular death and primary composite outcome.** Cumulative event rate of cardiovascular death (A+B) and primary composite outcome (C+D) at the end of follow-up and stratified by 1-year follow-up. BMS, bare-metal stent; CV, cardiovascular; DES, drug-eluting stent.

to reduce late loss–related coronary events may be an explanation for these findings. Although BMSs are considered to minimize the duration of DAPT in post-stent AF patients to reduce the bleeding risk [38], the new-generation DESs are preferred over BMSs in such patients at low bleeding risk [39]. Furthermore, in post-PCI AF patients, a modified HAS-BLED score was unable to predict bleeding events [40]. Therefore, the choice between BMSs and DESs should not be based on the bleeding score. While our patients have a systematically mandated 1-year DAPT regimen, except in patients with high bleeding risk, the gastrointestinal and major bleeding rates were comparable between groups in our study. Altogether, these observations suggest that stent type is important in determining outcomes of AF patients with first AMI requiring PCI.

**Table 3. Secondary outcome at the end of follow-up.**

| Outcome | Number of event (%) | | DES versus BMS HR (95% CI) | P |
|---|---|---|---|---|
| | BMS (n = 785) | DES (n = 436) | | |
| Any stroke | 67 (8.5) | 38 (8.7) | 0.93 (0.62–1.38) | 0.711 |
| Hemorrhagic stroke | 7 (0.9) | 2 (0.5) | 0.46 (0.10–2.22) | 0.334 |
| Unspecified stroke | 7 (0.9) | 2 (0.5) | 0.46 (0.10–2.21) | 0.332 |
| Gastrointestinal bleeding | 87 (11.1) | 40 (9.2) | 0.74 (0.51–1.07) | 0.107 |
| Major bleeding | 44 (5.6) | 23 (5.3) | 0.85 (0.51–1.41) | 0.527 |
| Major bleeding (include gastrointestinal bleeding) | 102 (13.0) | 46 (10.6) | 0.72 (0.51–1.02) | 0.061 |
| Heart failure admission | 99 (12.6) | 44 (10.1) | 0.72 (0.50–1.02) | 0.068 |

BMS, bare-metal stent; DES, drug-eluting stent; HR, hazard ratio; CI, confidence interval.

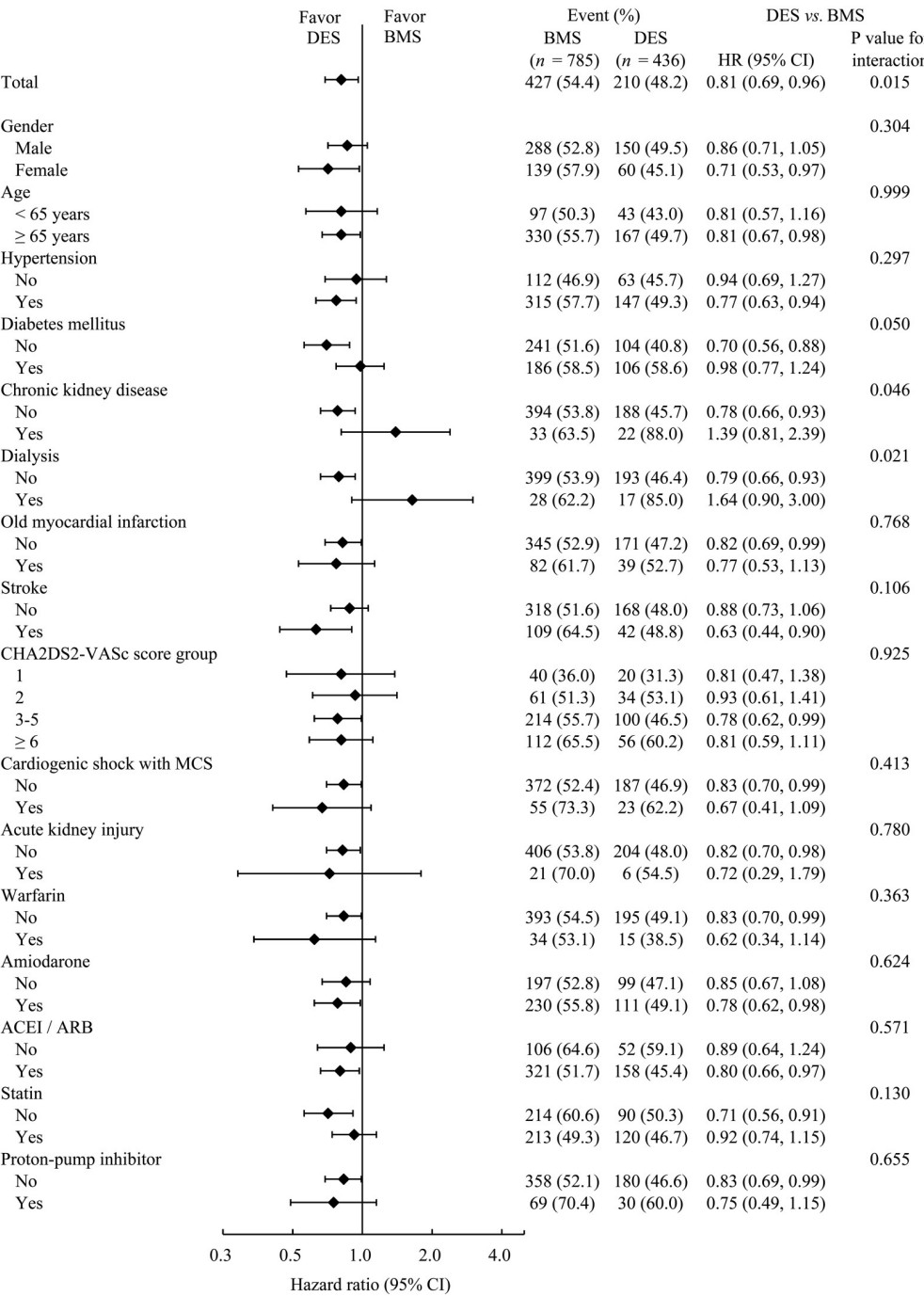

**Fig 3. Subgroup analyses.** Subgroup analyses for patient characteristics are shown with HRs and 95% CIs for the primary composite outcome at the end of follow-up. ACEI, angiotensin converting enzyme inhibitor; ARB, angiotensin-II receptor blocker; BMS, bare-metal stent; DES, drug-eluting stent; CI, confidence interval; HR, hazard ratio; MCS, mechanical circulation support.

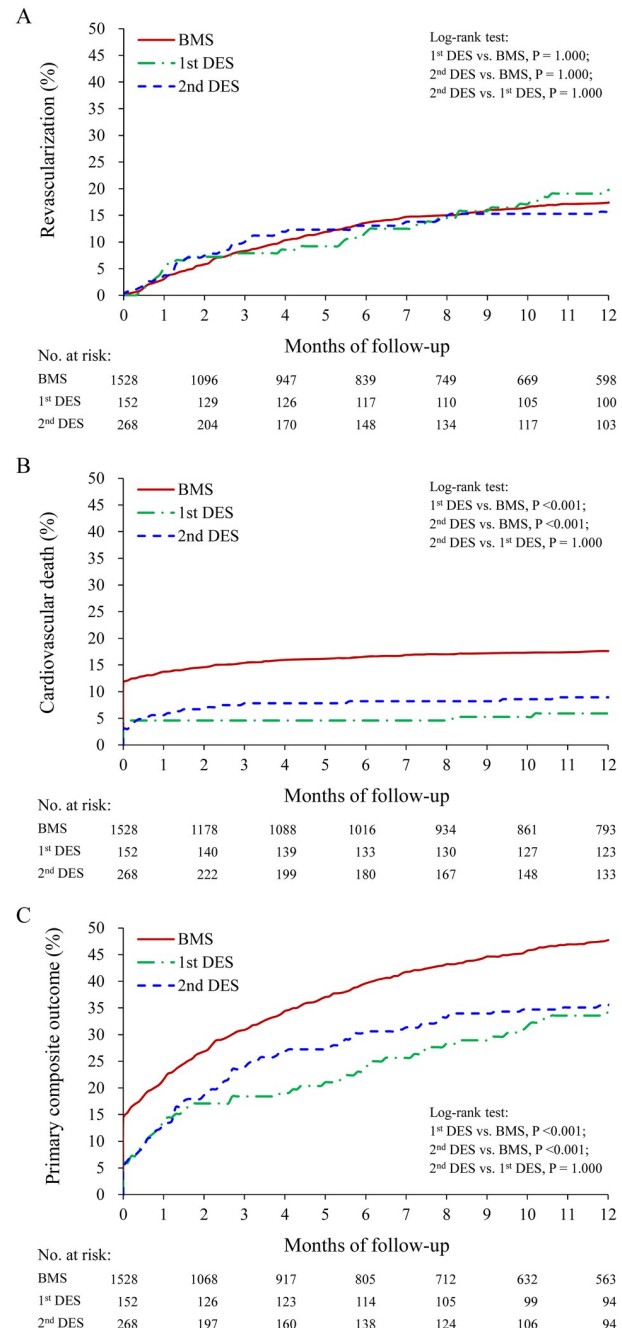

**Fig 4. Cumulative event rate at 1-year follow-up using BMS, first- or second-generation DESs.** Cumulative event rate of revascularization (A), cardiovascular death (B) and primary composite outcome (C) associated with different stent types at 1-year follow-up. BMS, bare-metal stent; CV, cardiovascular; DES, drug-eluting stent.

The use of DES in patients at high risk of bleeding or thrombosis has been recently studied. In ZEUS trial, among patients with uncertain ability to use DES with longer duration DAPT (median duration ~1 month) due to high bleeding, high thrombotic or low restenosis risk, second-generation zotarolimus-eluting stent was superior to BMS for clinical outcomes, including MI, stent thrombosis, and target lesion revascularization; bleeding risk was similar [41].

The LEADERS FREE trial showed that the composite primary safety end point of cardiac death, myocardial infarction, or stent thrombosis and the rate of the primary efficacy end point of clinically driven target-lesion revascularization.following biolimus A9 drug-coated stent implantation are lower than and superior to BMS in patients with high bleeding risk and who are able to take only 1 month of DAPT, while the latter strategy, driven by the need to minimize the risk of bleeding, is associated with a higher risk of restenosis and reintervention than that observed with the use of a DES [42]. Among elderly patients (age ≥75 years) undergoing PCI with a shorter duration of DAPT, the SENIOR trial showed that the use of a bioabsorbable polymer-DES resulted in lower adverse clinical event rates, including the composite primary endpoint of all-cause mortality, myocardial infarction, stroke, or revascularization, at 1 year compared with BMS [43]. This benefit was driven predominantly due to a lower risk of repeat revascularization with DES [43]. Although PCI patients are typically managed with DAPT, antiplatelet therapy alone has been shown to be inadequate for stroke prevention in AF [44]. Hence, post-PCI AF patients are increasingly treated with DES and oral anticoagulants [45]. Altogether, these data provide further evidence that shorter durations of DAPT may be feasible with DES in select patient populations, especially where the temptation has been to use BMS to minimize DAPT duration. [41].

Our finding that 9.7% of AMI patients had a history of AF is similar to a previous study [2]. The risk of ischemic stroke is approximately 1% in post-PCI AF patients [46]. In our study, the risk was even higher at approximately 5% at 1 year and 8% at the end of follow-up in both groups, suggesting additive effects of the intracoronary and systemic prothrombotic environment accompanying AF and AMI. In post-stent AF patients, coadministration of DAPT and oral anticoagulants raises the concern about bleeding risk. Among post-PCI AF patients, triple therapy has been found to have no association with a reduction in death, ischemic stroke, target vessel revascularization, MI, or major bleeding events versus DAPT [47]. Additionally, in our study, the use of DESs compared with BMSs was associated with a reduction of death in AF patients with first AMI and received 1-year DAPT. Furthermore, while the rate of ischemic or hemorrhagic stroke, myocardial infarction, revascularization and gastrointestinal bleeding was comparable between groups, the use of DESs was associated with a reduced incidence of death, cardiovascular death and primary composite outcome. Taken together, these studies suggest that in AF patients with first AMI who received stents and 1-year DAPT, changes in the design of stent platform and other unmeasured factors may contribute to the clinical benefit of DESs over BMSs.

In subgroup analyses, the effect of DESs was similar to that of BMSs for primary composite outcome in our AF patients who had diabetes mellitus, CKD, or who were on dialysis. While the current opinion of the impact of diabetes mellitus on the outcome after PCI remains speculative, among ST-segment elevation MI patients undergoing primary PCI, DES implantation does mitigate the deleterious effect of diabetes on target vessel revascularization after BMS [48]. Few studies are available on the safety and efficacy of DES in CKD patients because these patients are systematically excluded from major interventional cardiology trials. Shenoy et al. demonstrated that selective use of DESs was safe and effective in patients with CKD in the long term, with lower risk of the composite of major adverse cardiovascular events, defined as death, MI or target vessel revascularization, and similar risk of MI compared with the use of BMSs [49]. The discordance between our findings and those of Shenoy et al. [49] are likely due to the patient differences in the clinical and angiographic characteristics and potentially dissimilar unmeasured confounders. On the other hand, the use of DESs compared to BMSs reduced the risk of all-cause mortality at 17 months in the study by Zhang et al. [50] and did not reduce mortality at 4 years in the study by Appleby et al [51]. However, they did not use any statistical designs (matching, covariate adjustment, or propensity-based adjustment) to

adjust for differences between the DES and BMS patients. Of note, Appleby et al. found a significant survival benefit from DESs compared to BMSs in the first year after treatment (P = 0.002), with catch-up at 2 years (P = 0.057) [51].

Controversy remains regarding the efficacy of DES compared to BMS implantation in dialysis patients. While a study has demonstrated a reduced risk of recurrent MI, cardiovascular death, and all-cause mortality with the use of DESs compared with BMSs [52], none of the previous studies has focused on AF patients on dialysis with first AMI as the study population. Dialysis is reported to be an independent predictor of late catch-up phenomenon [53]. Whether the loss of DES treatment effects in AF patients with first AMI who had diabetes mellitus, CKD or dialysis is related to late catch-up phenomenon needs further studies.

Our study demonstrated a significantly lower cumulative incidence of cardiovascular death and primary composite outcome in the DES than BMS group within 1 year but a similar cumulative incidence from 1 year to end of the follow-up, which is similar to that of other studies [54]. Although late complications such as very late stent thrombosis and late catch-up phenomenon in patients after DES implantation have been previously reported [53], it is hard to define the exact rate of very late stent thrombosis because of the lack of data in the NHIRD. However, our results suggest a possible contribution of late catch-up phenomenon to cardiovascular death and primary composite outcome in AF patients with first AMI who received DES compared with those who received BMS. Further studies are warranted to elucidate this issue.

Our study has several limitations. First, our retrospective case-control study has a lower level of evidence than a prospective study. The common confounders of patient information, such as pre-infarction angina, family history of cardiovascular disease, smoking, body mass index, lipid profile, residual renal function, and dialyzer membrane type, were lacking in the NHIRD. Nonetheless, a wide range of variables associated with outcomes were included to match our 2 study groups. Second, although we analyzed repeat revascularizations, we could not determine the target vessel, target lesion revascularization, and different types of coronary revascularization. Third, the lack of clinical information in the claims database regarding stent thrombosis, angiographic characteristics, and lesion classification did not allow for a more detailed analysis. Finally, the available Taiwan NHIRD in our study included only inpatient claims data with no information of medications from the outpatient claims data and pharmacy claims data. Furthermore, although the DOAC has been covered by Taiwan's NHI system since 2012, our study period was from 1997 to 2011. Therefore, the use of oral anticoagulants was infrequent in our study. Nonetheless, the examination of large national databases is valuable, depending on an appropriate hypothesis, proper study design, and careful analysis of the results.

## Conclusions

Due to the complexity of treating AF patients who experience a first AMI requiring PCI, much effort should be made to come up with cost-effective treatment strategies. From the perspective of implanting a stent, the implantation of DESs compared with BMSs leads to a lower risk of primary composite outcome such as ischemic stroke, MI, revascularization, and death despite 1-year DAPT in both groups, suggesting differential prognostic impacts for DESs and BMSs. However, the effect is less apparent in patients with diabetes, CKD or on dialysis.

## Supporting information

**S1 Table. ICD-9-CM code.**
(DOC)

## Acknowledgments

We thank Alfred Hsing-Fen Lin and Zoe Ya-Jhu Syu for their assistance in statistical analysis.

## Author Contributions

**Conceptualization:** Nen-Chung Chang, Ming-Yow Hung.

**Data curation:** Tien-Hsing Chen.

**Formal analysis:** Ming-Yow Hung.

**Funding acquisition:** Nen-Chung Chang, Ming-Yow Hung.

**Investigation:** Patrick Hu, Chi-Tai Yeh.

**Methodology:** Nen-Chung Chang, Ming-Yow Hung.

**Project administration:** Chi-Tai Yeh, Ming-Yow Hung.

**Resources:** Ming-Yow Hung.

**Software:** Tien-Hsing Chen, Ming-Yow Hung.

**Supervision:** Tien-Hsing Chen, Chun-Tai Mao, Ming-Jui Hung.

**Validation:** Nen-Chung Chang, Tien-Hsing Chen, Chun-Tai Mao, Ming-Jui Hung, Ming-Yow Hung.

**Visualization:** Ming-Yow Hung.

**Writing – original draft:** Nen-Chung Chang, Ming-Yow Hung.

**Writing – review & editing:** Ming-Yow Hung.

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
