## [Decision Letter · Decision Letter 0]

24 Oct 2019

PONE-D-19-23376

Drug-eluting versus bare-metal stents for first myocardial infarction in patients with atrial fibrillation: a nationwide population-based cohort study

PLOS ONE

Dear Ming-Yow Hung

Thank you for submitting your manuscript to PLOS ONE. After careful consideration, we feel that it has merit but does not fully meet PLOS ONE’s publication criteria as it currently stands. Therefore, we invite you to submit a revised version of the manuscript that addresses the points raised during the review process.

We would appreciate receiving your revised manuscript by Dec 08 2019 11:59PM. To enhance the reproducibility of your results, we recommend that if applicable you deposit your laboratory protocols in protocols.io, where a protocol can be assigned its own identifier (DOI) such that it can be cited independently in the future. For instructions see: http://journals.plos.org/plosone/s/submission-guidelines#loc-laboratory-protocols

We look forward to receiving your revised manuscript.

Kind regards,

Giuseppe Coppola

Academic Editor

PLOS ONE

Additional Editor Comments (if provided):

Congratulations for your manuscript and thank you for considering Plos One.

Your paper underwent two separate revisions. Both reviewers agree about **minor revision**. Their suggestione are well explained in the comments.

We look forward to see your revised version.

Reviewers' comments:

Reviewer's Responses to Questions

**Comments to the Author**

1. Is the manuscript technically sound, and do the data support the conclusions?

Reviewer #1: Yes

Reviewer #2: Yes

2. Has the statistical analysis been performed appropriately and rigorously? 

Reviewer #1: Yes

Reviewer #2: Yes

3. Have the authors made all data underlying the findings in their manuscript fully available?

Reviewer #1: Yes

Reviewer #2: Yes

4. Is the manuscript presented in an intelligible fashion and written in standard English?

Reviewer #1: Yes

Reviewer #2: Yes

5. Review Comments to the Author

Reviewer #1: Introduction: the quoted guidelines are outdated; the Authors should review the entire section according to the last iteration of the ACC/AHA and ESC guidelines, both about type of stent and about antithrombotic regimen; for completeness, they should put their study (enrolling started more than 10 years ago) in context to the evidences of that era.

Discussion: the Authors should discuss their study in the light of three major RCTs comparing DES and BMS for patients considered at high risk of bleeding, such as ZEUS (Valgimigli et al.), LEADES-FREE (Urban et al.) and SENIOR (Varenne et al.) , and also of observational studies reporting management of patients with AF and coronary stents (see Potter at al, Clin Cardiol. 2018 Apr;41(4):470-475. doi: 10.1002/clc.22898).

Reviewer #2: The authors deals with a very important issue, that is to say use of antiplatelet therapy in patients already on anticoagulants for AF. The paper is well written and the statistical analysis solid, thus the conclusions are reliable. I only have some minor questions:

1. In Table 1 patients are divided in categories according to their CHA2DS-VASc score, and score 1 and 2 are included in the some category. In my opinion pts with a CHA2DS2-VASc score = 1 should be separated from pts with a score of 2, because they could even not receive an anticoagulant

2. How can pts have a score of 0, if they have at least cardiovascular disease?

3. What about pts not on warfarin? Only about 9% of pts are on warfarin: and the other ones? Are they on DOAC? Which type at which dose: full or reduced?

6. PLOS authors have the option to publish the peer review history of their article (what does this mean?). If published, this will include your full peer review and any attached files.

Reviewer #1: No

Reviewer #2: Yes: Giosue Mascioli, MD; FESC, FEHRA, FAIAC

---

## [Author Response · Author response to Decision Letter 0]

15 Dec 2019

Response to reviewers

 We would like to extend our thanks to the reviewers for the constructive comments regarding our manuscript, entitled, “Drug-eluting versus bare-metal stents for first myocardial infarction in patients with atrial fibrillation: A nationwide population-based cohort study” (Manuscript number PONE-D-19-23376). We have revised the manuscript point by point in accordance with the reviewer’s suggestions, as indicated by the itemized responses below.

Reviewer #1:

1. Introduction: the quoted guidelines are outdated; the Authors should review the entire section according to the last iteration of the ACC/AHA and ESC guidelines, both about type of stent and about antithrombotic regimen; for completeness, they should put their study (enrolling started more than 10 years ago) in context to the evidences of that era.

Response to reviewer: 

We appreciate the reviewer’s comments. We have reviewed the entire section according to the last iteration of the ACC/AHA and ESC guidelines, both about type of stent and about antithrombotic regimen. We also have described the development of evidence over time through our study period. Guidelines and reviews of the literature have been added to the “Introduction” sections in this revised manuscript (Page 3, lines 65-68, Page 4, lines 69-84 and 89-91 and Page 5, lines 92, 94 and 97-101). Thank you for your comment.

2. Discussion: the Authors should discuss their study in the light of three major RCTs comparing DES and BMS for patients considered at high risk of bleeding, such as ZEUS (Valgimigli et al.), LEADES-FREE (Urban et al.) and SENIOR (Varenne et al.), and also of observational studies reporting management of patients with AF and coronary stents (see Potter at al, Clin Cardiol. 2018 Apr;41(4):470-475. doi: 10.1002/clc.22898).

Response to reviewer: 

We appreciate the reviewer’s comments. As the reviewer suggested, we have added the detailed reviews of the 3 major RCTs and 1 observational study regarding the management of patients with AF and coronary stents to the “Discussion” section in this revised manuscript (Page 20, lines 334-338, Page 21, lines 359-370 and Page 22, lines 371-380). Thank you for your comment.

Reviewer #2:

1. In Table 1 patients are divided in categories according to their CHA2DS-VASc score, and score 1 and 2 are included in the same category. In my opinion pts with a CHA2DS2-VASc score = 1 should be separated from pts with a score of 2, because they could even not receive an anticoagulant.

Response to reviewer: 

We appreciate the reviewer’s comments. We have modified and improved the data presentation of the subgroup analysis in Table 1 and Figure 3. Thank you for your comment.

2. How can pts have a score of 0, if they have at least cardiovascular disease?

Response to reviewer: 

We appreciate the reviewer’s comments. While all patients included in this study should have at least a CHA2DS2-VASc score of 1, the reason why some patients had a score of 0 is that we defined the coronary artery disease by using ICD-9-CM diagnosis prior to the index admission. We apologize for this coding error, and have redefined the coronary artery disease using ICD-9-CM diagnosis “during” or prior to the index admission. We have modified and improved the data presentation of the subgroup analysis in Table 1 and Figure 3. Thanks for your reminder and comment.

3. What about pts not on warfarin? Only about 9% of pts are on warfarin: and the other ones? Are they on DOAC? Which type at which dose: full or reduced?

Response to reviewer: 

We appreciate the reviewer’s comments. The available Taiwan NHIRD in our study included only inpatient claims data with no information of medications from the outpatient claims data and pharmacy claims data. Furthermore, although the DOAC has been covered by Taiwan’s NHI system since 2012, our study period was from 1997 to 2011. Therefore, the use of oral anticoagulants was infrequent in our study. We have added the above statement to the limitation section in this revised manuscript (Page 25, lines 442-446). Thank you for your comment.

List of changes

The page and reference numbers in this list are those in the revised and re-submitted manuscript.

All of the above revisions are highlighted with underlines and red color in the revised manuscript. Thank you very much for your recommendations.

1. Page 3, lines 65-68, Page 4, lines 69-84 and 89-91 and Page 5, lines 92, 94 and 97-101: As the reviewer suggested, guidelines and reviews of the literature were added.

2. Page 20, lines 334-338, Page 21, lines 359-370 and Page 22, lines 371-380: As the reviewer suggested, the detailed reviews of the 3 major RCTs and 1 observational study regarding the management of patients with AF and coronary stents were added. 

3. Page 25, lines 442-446: As the reviewer suggested, the reasons why the use of oral anticoagulants was infrequent in our study were added.

4. Page 26, line 479: The author contribution for revising this manuscript was added.

5. Page 28, 29, 30, 31, 33 and 35: The reference number was sequentially changed as follows: 11 to 9, 8 to 10, 9 to 17, 10 to 18, 12 to 21, 13 to 22, 14 to 23, 15 to 24, 16 to 25, 17 to 26, 18 to 27, 19 to 28, 20 to 29, 21 to 30, 22 to 31, 23 to 32, 24 to 34, 25 to 35, 26 to 36, 27 to 37, 28 to 38, 29 to 39, 30 to 40, 31 to 46, 32 to 47, 33 to 48, 34 to 49, 35 to 50, 36 to 51, 37 to 52, 38 to 53, 39 to 54.

6. Page 28: One reference (J Am Coll Cardiol. 2006; 48:854-906.) was added as reference 8.

7. Pages 29-30: 6 references (Lancet. 2013; 381:1107-15.; Thromb Res. 2015; 135:26-30.; Circulation. 2019; 140:e125-e151.; N Engl J Med. 2016; 375:2423-34.; N Engl J Med. 2017; 377:1513-24.; Circulation. 2018; 138:527-36.) were added as reference 11-16.

8. Pages 30-31: 2 references (J Am Coll Cardiol. 2011; 58:e44-122.; Circ Cardiovasc Interv. 2016; 9. pii: e004395.) were added as reference 19, 20.

9. Page 33: One reference (Circulation. 2008; 117:261-95.) was added as reference 33.

10. Pages 35: 5 references (J Am Coll Cardiol. 2015; 65:805-15.; N Engl J Med. 2015; 373:2038-47.; Lancet. 2018; 391:41-50.; Ann Intern Med. 2007; 146:857-67.; Clin Cardiol. 2018; 41:470-75.) were added as reference 41-45.

---

## [Editor Report · Decision Letter 1]

23 Dec 2019

Drug-eluting versus bare-metal stents for first myocardial infarction in patients with atrial fibrillation: a nationwide population-based cohort study

PONE-D-19-23376R1

Dear Dr. Ming-Yow Hung,

We are pleased to inform you that your manuscript has been judged scientifically suitable for publication and will be formally accepted for publication once it complies with all outstanding technical requirements.

With kind regards,

Giuseppe Coppola

Academic Editor

PLOS ONE

---

## [Editor Report · Acceptance letter]

31 Dec 2019

PONE-D-19-23376R1 

Drug-eluting versus bare-metal stents for first myocardial infarction in patients with atrial fibrillation: a nationwide population-based cohort study 

Dear Dr. Hung:

I am pleased to inform you that your manuscript has been deemed suitable for publication in PLOS ONE. Congratulations! Your manuscript is now with our production department. 

With kind regards,

on behalf of

Dr. Giuseppe Coppola 

Academic Editor

PLOS ONE